# Risk factors for lactation mastitis in China: A systematic review and meta-analysis

**Bao-Yong Lai**[1], **Bo-Wen Yu**[1], **Ai-Jing Chu**[1], **Shi-Bing Liang**[2], **Li-Yan Jia**[3], **Jian-Ping Liu**[2], **Ying-Yi Fan**[1]☯*, **Xiao-Hua Pei**[1,4]☯*

**1** Third Affiliated Hospital of Beijing University of Chinese Medicine, Beijing, China, **2** Centre for Evidence-Based Chinese Medicine, Beijing University of Chinese Medicine, Beijing, China, **3** School of Traditional Chinese Medicine, Beijing University of Chinese Medicine, Beijing, China, **4** The Xiamen Hospital of Beijing Universality of Chinese Medicine, Xiamen, China

☯ These authors contributed equally to this work.
* pxh_127@163.com (XHP); fan38898901@126.com (YYF)

**Data Availability Statement:** All relevant data are within the manuscript and its Supporting information files.

**Funding:** This work was supported by the Project and Study on clinical evaluation of traditional

## Abstract

### Background

Lactation mastitis (LM) affects approximately 3% to 33% of postpartum women and the risk factors of LM have been extensively studied. However, some results in the literature reports are still not conclusive due to the complexity of LM etiology and variation in the populations. To provide nationally representative evidence of the well-accepted risk factors for LM in China, this study was aimed to systematically summary the risk factors for LM among Chinese women and to determine the effect size of individual risk factor.

### Material and methods

Six major Chinses and English electronic literature databases (PubMed, Web of Science, Chinese Biomedical Literature Database, China National Knowledge Infrastructure, Wan fang Database and China Science Technology Journal Database) were searched from their inception to December 5st, 2020. Two authors extracted data and assessed the quality of included trials, independently. The strength of the association was summarized using the odds ratio (OR) with 95% confidence intervals (CI). The population attributable risk (PAR) percent was calculated for significant risk factors.

### Results

Fourteen studies involving 8032 participants were included. A total of 18 potential risk factors were eventually evaluated. Significant risk factors for LM included improper milking method (OR 6.79, 95%CI 3.45–13.34; PAR 59.14%), repeated milk stasis (OR 6.23, 95%CI 4.17–9.30; PAR 49.75%), the first six months postpartum (OR 5.11, 95%CI 2.66–9.82; PAR 65.93%), postpartum rest time less than 3 months (OR 4.71, 95%CI 3.92–5.65; PAR 56.95%), abnormal nipple or crater nipple (OR 3.94, 95%CI 2.34–6.63; PAR 42.05%), breast trauma (OR 3.07, 95%CI 2.17–4.33; PAR 15.98%), improper breastfeeding posture (OR 2.47, 95%CI 2.09–2.92; PAR 26.52%), postpartum prone sleeping position (OR 2.46, 95%CI 1.58–3.84; PAR 17.42%), little or no nipple cleaning (OR 2.05, 95%CI 1.58–2.65;

Chinese medicine in reducing the use of antibiotics during the treatment of acute suppurative mastitis in the form of funds awarded to XHP (No. Shou Fa 2018-7032). The funder had no role in study design, data collection and analysis, decision to publish, or preparation of the manuscript.

**Competing interests:** The authors have declared that no competing interests exist.

PAR 24.73%), primipara (OR 1.73, 95%CI 1.25–2.41; PAR 32.62%), low education level (OR 1.63, 95%CI 1.09–2.43; PAR 23.29%), cesarean section (OR 1.51, 95%CI 1.26–1.81; PAR 18.61%), breast massage experience of non-medical staff (OR 1.51, 95%CI 1.25–1.82; PAR 15.31%) and postpartum mood disorders (OR 1.47, 95%CI 1.06–2.02; PAR 21.27%).

## Conclusions

This review specified several important risk factors for LM in China. In particular, the incidence of LM can be reduced by controlling some of the modifiable risk factors such as improper breastfeeding posture, improper milking method, repeated milk stasis, nipple cleaning, breast massage experience of non-medical staff and postpartum sleeping posture.

## 1 Introduction

Lactation mastitis (LM) is one of the most common breast disorders experienced by postpartum women [1]. It is clinically characterized by a red, swollen, hot and tender area of the breast generally accompanied by fever, headache, and other influenza-like symptoms [2]. The incidence of LM is between 3% to 33% due to variation in the populations and follow-up in the postpartum period [3, 4]. LM occurs frequently in the first six to eight weeks of postpartum but it can also occur at any time during breastfeeding [5]. In addition, previous studies have shown that mismanagement or incorrect breast care can lead to the development of LM into severe cases (such as breast abscess or sepsis), which would directly lead to the cessation of normal breastfeeding [5, 6].

The World Health Organization (WHO) or international guidelines highly recommends that infants are exclusively breastfed for the first six months of life and continue breastfeeding for up to two years of age or older, because breastfeeding can provide the best nutritional start for infant growth [7, 8] and it has beneficial effects on the health outcomes of both infants and mothers [9]. Unfortunately, it was reported that one of the main causes directly inducing breastfeeding failure were LM and its related discomfort [9–11]. Given the beneficial effects of breastfeeding and China having the highest population in the world and Asia, it is of concern that previous surveys in China reported that the breastfeeding rate of infants aged 1–2 months ranged from 59.4% to 66.5% and the rate of exclusive breastfeeding was only 15.8% for infants below six months old [10, 12, 13]. Therefore, it is of great significance to explore the risk factors associated with LM and in order to prolong lactation.

Some researchers had studied the risk factors related to LM among Chinese women and the results revealed that some risk factors involving sociodemographic characteristics, breastfeeding behaviors and psychological mood were associated with LM [12, 14, 15]. However, some results in the literature reports on this topic are still not conclusive (such as the breastfeeding behaviors and puerperium characteristics) due to the complexity of LM etiology [12, 15, 16]. Another recent systematic review [17] published in 2020 summarized the evidence on risk factors for LM in the word. However, the effect size of risk factors was not finally pooled due to methodological differences in these studies. Therefore, it is critical and necessary for lactating mothers or practitioners to detect and avoid the high-risk factors associated with LM and a clearer understanding of the risk factors for LM is needed. To provide nationally

representative evidence of the well-accepted risk factors for LM in China, we performed this systematic review to determine and clarify the significant risk factors related to LM among Chinese women. Furthermore, to estimate the potential impact of these factors on LM at the population level, the population attributable risk (PAR) percent was calculated where possible.

## 2 Methods

A systematic review and meta-analysis of relevant studies was conducted and reported, following the PRISMA recommendations [18]. The protocol of this review has been registered at PROSPERO (CRD42020186674).

### 2.1 Eligibility/exclusion criteria

The following criteria were used to identify relevant studies: (1) This review included case-control studies, cohort studies, cross-sectional studies, and randomized controlled trials (RCTs) to explore the risk factors associated with LM; (2) All considered participants were Chinese women. If the studies are mixed populations, data from Chinese women could be analyzed separately, regardless of their age or race; and (3) English and Chinese language publications.

Studies were excluded from the analysis: (1) data could not be extracted; (2) Studies where the outcome was not clearly stated and (3) Studies that included duplicate data.

### 2.2 Search strategy

We systematically searched PubMed, Web of Science, Chinese Biomedical Literature Database (SinoMed), China National Knowledge Infrastructure (CNKI), Wan fang Database, and China Science Technology Journal Database (VIP) from their inception to December 5st, 2020. The following search terms were used: (mastitis [MeSH Terms] or acute mastitis) and (risk factor [MeSH Terms] or risk factors or influence factors or factor analysis) and (Chinese or China). S3 File outlined the detailed search strategy of PubMed.

### 2.3 Study selection and data extraction

Two authors independently selected the studies and extracted the detailed data of the eligible studies. The items for data extraction were first authors, year of publication, study type, study setting, the detailed information of methodology, characteristics of participants, sample size the data of risk factors associated with LM and response rate. Any discrepancies regarding study selection and data extraction were resolved through consensus and arbitrated by the third author if necessary.

### 2.4 Quality assessment

The quality of the case-control study and cohort study was respectively assessed according to the criteria of the Newcastle-Ottawa Scale (NOS) [19], and the quality of cross-sectional study was evaluated using the modified NOS [20, 21]. The "star" scoring system of NOS was used during the evaluation process and a star was described as an appropriate entry, with each star representing one point. Studies with a high score indicated a good quality study, those with a score of six or greater were considered as acceptable quality, and those with a total score >7 were considered high-quality studies [22, 23]. We evaluated the quality of RCTs by using the Cochrane risk of bias tool [24]. Two authors independently made judgements about Quality assessment. Any disagreement was resolved by discussion with a third author.

## 2.5 Statistical analysis

We used RevMan5.3 and Stata14.0 to perform statistical analysis, binary data were summarized using odds ratio (OR) with their 95% confidence intervals (CI). We assessed statistical heterogeneity by using the $I^2$ statistics test and Q chi-squared test. When $I^2 > 50\%$ and Q chi-squared test result $< 0.1$, it shows that there is significant statistical heterogeneity among the trials, and the random effect model was adopted. Otherwise, it shows that there is no obvious statistical heterogeneity among the trials, and the fixed effect model was used [25]. Sensitivity analysis was performed when possible to test the robustness of the results. The PAR percent were calculated to indicate the proportion of cases that can be attributed to each risk factor according to the following formula [26]. $PAR = \frac{Pe(OR-1)}{1+Pe(OR-1)}$. The PAR percent is calculated using the pooled OR for each risk factor and is estimated based on the identified meta-analysis. '$Pe$' is the prevalence of exposure in the population.

The fail-safe number (Nfs) was calculated to measure publication bias according to the following formula. Nfs0.05 = $(\Sigma Z/1.64)^2$-K, Nfs0.01 = $(\Sigma Z /2.33)^2$-K, the K in the formula is the number of selected studies. The larger the value of Nfs, the smaller the bias [27]. Additionally, the Nfs value was used to estimate the strength of the evidence by calculating the number of negative studies required to nullify current results. Furthermore, Egger's linear regression tests were performed to further evaluate publication bias.

# 3 Results

## 3.1 The selection of study

A total of 265 related articles were obtained from 6 databases. First duplicates were excluded, and then 127 articles were excluded by reading the title and abstract. Full texts of 25 articles were screened according to the eligibility criteria. Finally, fourteen articles met inclusion criteria and were included for analysis. The selection process was showed in Fig 1.

## 3.2 Study characteristics and quality assessment of included studies

In total, fourteen studies were included, involving a combined total of 8032 participants. Eleven of the studies [28–38] were case-control studies (case groups were patients with mastitis in lactation and control groups were healthy women with previous breastfeeding experience). One [39] was a prospective cohort study (mothers who had delivered healthy babies at seven health facilities were recruited to participate in a face-to-face interview before discharge and then follow-up interviews were conducted at 1, 3, and 6 months postpartum by telephone. Forty-two mothers reported at least one episode of mastitis and 628 breastfeeding mothers were with no mastitis during the first 6 months postpartum). The rest two [40, 41] were cross-sectional studies. All the mothers who participated in the included studies were recruited at local hospitals or health facilities and a questionnaire was developed to collect data on general sociodemographic, psychosocial and puerperium characteristics, except for one study [28] only involved a retrospective review of medical records. Eleven of the included studies were published in Chinese [29–36, 38, 40, 41] and three studies were published in English [28, 37, 39]. The basic information of included studies was shown in Table 1.

The overall quality of the included studies was acceptable. Five of the included studies were of high quality according to the NOS criteria. Detailed information of quality assessment was shown in S1–S3 Tables.

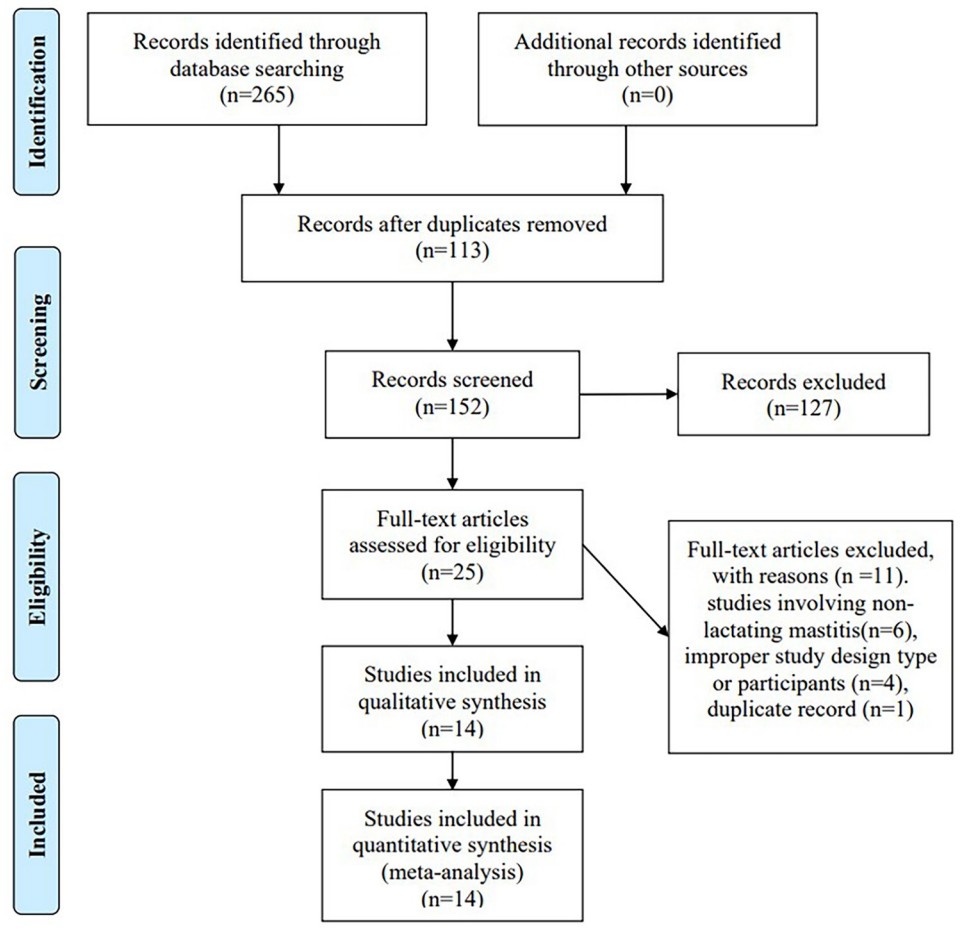

**Fig 1. Flow diagram of the literature search and selection processes.**

## 3.3 Results of meta-analysis

More than two studies involving the same defined risk factor for LM were summarized in the meta-analysis. A total of 18 potential risk factors were identified, of which 6 were classified as risk factors related to puerperium behaviors and characteristics (improper breastfeeding posture, repeated milk stasis, improper milking method, little or no nipple cleaning, each breastfeeding duration>0.5h and sucking manners of infants). Eight of them were classified as risk factors related to maternal characteristics (cesarean section, breast massage experience of non-medical staff, history of diabetes, history of mastitis, abnormal nipple or crater nipple, primipara, breast trauma and low education level). Following four factors were categorized as risk factors related to the postpartum period (postpartum rest time, the first six months postpartum, postpartum sleeping posture and postpartum mood disorders).

**3.3.1 Risk factors related to puerperium behaviors and characteristics.** There was no significant heterogeneity ($I^2 \leq 50\%$) among the following risk factors: improper breastfeeding posture (P = 0.79, $I^2$ = 0%) and repeated milk stasis (P = 0.79, $I^2$ = 0%), these data were pooled using the fixed effect model. The pooled risks showed that improper breastfeeding posture or laid-back breastfeeding [30, 33, 34] (OR 2.47, 95%CI [2.09, 2.92]) and repeated milk stasis [31, 32, 38] (OR 6.23, 95%CI [4.17, 9.30]) were identified as significant risk factors for LM. The forest plots are shown in Fig 2.

**Table 1. Basic information of the included studies.**

| Study ID | research design type | province/area | study duration | sample size | | age(years) | | NOS |
|---|---|---|---|---|---|---|---|---|
| Zhong HY 2018 [28] | case control study | Shandong | 2013–2017 | case:63 | control:262 | case:NR | control:NR | 6 |
| He XP 2013 [29] | case control study | Beijing | 2011–2012 | case:237 | control:237 | case:29.9±3.0 | control:27.3±3.6 | 8 |
| Pu YN 2017 [30] | case control study | Zhejiang | 2011–2015 | case:1000 | control:1000 | case:NR | control:NR | 6 |
| Li JX 2019 [31] | case control study | Guangdong | 2015–2018 | case:135 | control:135 | case:NR | control:NR | 6 |
| Wang HM 2016 [32] | case control study | Fujian | 2015–2016 | case:241 | control:241 | case:27.5±5.63 | control:31.2±5.0 | 8 |
| Cheng MH 2014 [33] | case control study | Guangdong | 2013–2013 | case:100 | control:100 | case:NR | control:NR | 6 |
| Zhai HL 2017 [34] | case control study | Henan | 2014–2017 | case:224 | control:224 | case:28.61±3.05 | control:29.24±3.19 | 7 |
| Gao X 2015 [35] | case control study | Chongqing | 2013–2014 | case:100 | control:100 | case:29.33 ±9.2 | control:29.12±8.35 | 7 |
| Chen XG 2016 [36] | case control study | Guangdong | 2010–2014 | case:313 | control:267 | case:NR | control:NR | 6 |
| Yin YS 2020 [37] | case control study | Shandong | 2016–2017 | case:652 | control:581 | case:29.89±3.37 | control:30.26±3.78 | 8 |
| Hu XC 2020 [38] | case control study | Tianjin | 2018–2019 | case:52 | control:184 | case:NR | control:NR | 6 |
| Li T 2014 [39] | prospective cohort study | Sichuan | 2010–2011 | 670 (mastitis:42, no mastitis:628) | | 24 | | 8 |
| Xia HL 2011 [40] | cross sectional study | Jiangsu | 2006–2010 | 846 | | NR | | 8 |
| Wang XL2018 [41] | cross sectional study | Shanxi | 2017–2018 | 68 | | 29.14±1.36 | | 6 |

Note: case: case group, control: control group, NR: Not reported, NOS (score): Newcastle-Ottawa Scale.

There was obvious heterogeneity among the following risk factors ($I^2>50\%$): improper milking method (P = 0.07, $I^2 = 69\%$), little or no nipple cleaning before breastfeeding (P = 0.007, $I^2 = 70\%$), each breastfeeding duration>0.5 h (P<0.0001, $I^2 = 83\%$) and nipple sucking (P<0.0001, $I^2 = 98\%$), the random effect model was used for the analysis of these variables. The pooled risks showed that the improper milking method [30, 31] (OR 6.79, 95%CI [3.45, 13.34]) and little or no nipple cleaning before breastfeeding [29–32, 34, 35, 37–39] (OR 2.05, 95%CI [1.58, 2.65]) were identified as the important risk factors for LM. No statistically significant differences were found for the other two risk factors: each breastfeeding duration>0.5 h [29, 31, 32, 34, 35, 39] (OR 0.77, 95%CI [0.48, 1.24]) and nipple sucking [29–31, 33–38] (OR 0.90, 95%CI [0.29, 2.72]). The forest plots are shown in Fig 3.

**3.3.2 Risk factors related to maternal characteristics.** There was no significant heterogeneity among the following risk factors ($I^2≤50\%$): cesarean section (P = 0.16, $I^2 = 41\%$), breast massage experience of non-medical staff (P = 0.15, $I^2 = 47\%$), history of diabetes (P = 0.16, $I^2 = 46\%$) and history of mastitis (P = 0.8, $I^2 = 0\%$), the data of these factors were pooled using the fixed effect model. The pooled risks showed that cesarean section [31, 36, 37, 39] (OR 1.51,

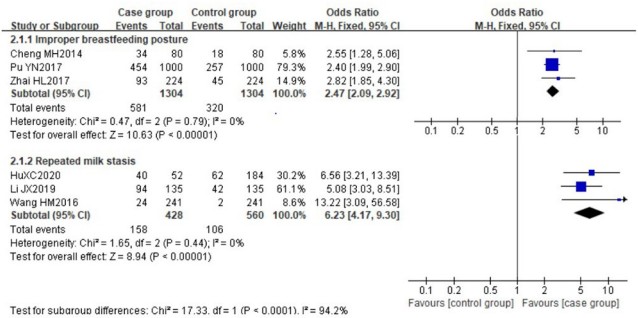

**Fig 2. Forest plot comparing the pooled risks of improper breastfeeding posture and repeated milk stasis.**

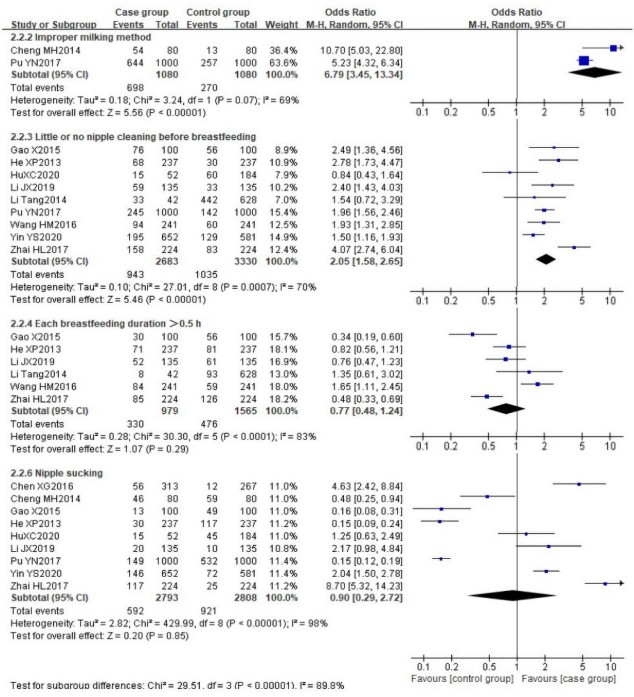

**Fig 3. Forest plot comparing the pooled risks of improper milking method, little or no nipple cleaning before breastfeeding, each breastfeeding duration>0.5 h and nipple sucking.**

95%CI [1.26, 1.81]), breast massage experience of non-medical staff [31, 34, 37] (OR 1.51, 95% CI [1.25, 1.82]), history of diabetes [28, 31, 32] (OR 2.26, 95%CI [1.43, 3.58]) and history of mastitis [32, 36, 37] (OR 2.36, 95%CI [1.84, 3.04]) were identified as the significant risk factors for LM. The forest plots are shown in Fig 4.

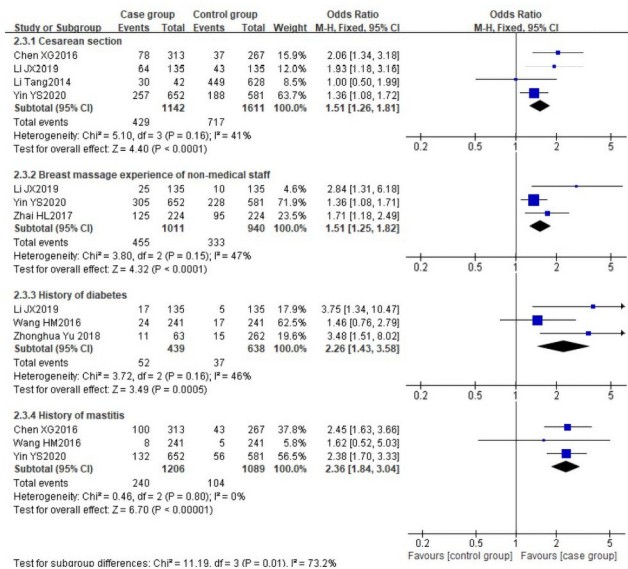

**Fig 4. Forest plot comparing the pooled risks of cesarean section, breast massage experience of non-medical staff, history of diabetes and history of mastitis.**

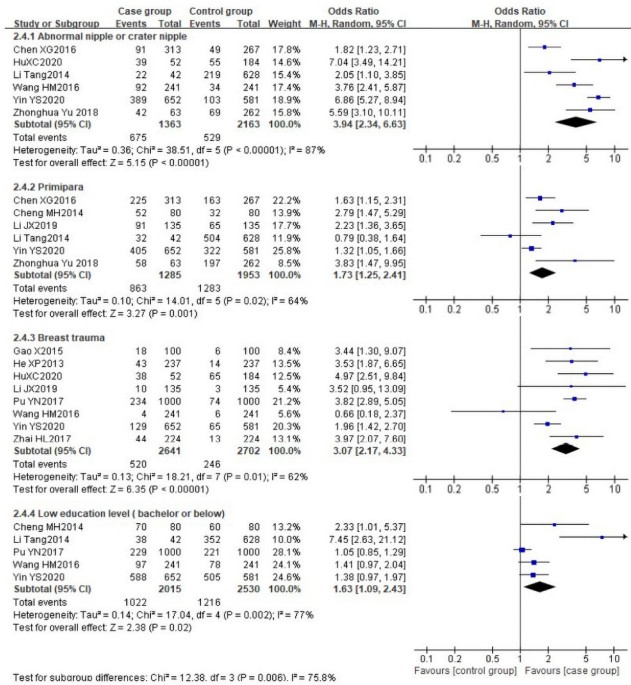

**Fig 5. Forest plot comparing the pooled risks of abnormal nipple or crater nipple, primipara, breast trauma and low education level.**

There was obvious heterogeneity among the following risk factors ($I^2$>50%): abnormal nipple or crater nipple (P<0.00001, $I^2$ = 87%), primipara (P = 0.02, $I^2$ = 64%), breast trauma (P = 0.01, $I^2$ = 62%) and low education level (P = 0.002, $I^2$ = 77%), the data of these factors were pooled using the random effect model. The pooled risks showed that abnormal nipple or crater nipple [28, 32, 36–39] (OR 3.94, 95%CI [2.34, 6.63]), primipara [28, 31, 33, 36, 37, 39] (OR 1.73, 95%CI [1.25, 2.41]), breast trauma [29–32, 34, 35, 37, 38] (OR 3.07, 95%CI [2.17, 4.33]) and low education level (bachelor below) [30, 32, 33, 37, 39] (OR 1.63, 95%CI [1.09, 2.43]) were identified as the significant risk factors for LM. The forest plots are shown in Fig 5. Similarly, the result from one cross-sectional study [40] involving 864 participants reported that primipara (OR 3.46, 95%CI [1.04, 11.46]) and a mother with low education level (high school or below) (OR 2.2, 95%CI [1.11, 4.35]) experienced a higher risk of LM.

**3.3.3 Risk factors related to postpartum period.** As for the relationship between rest time of postpartum women and LM, there was no significant heterogeneity ($I^2$≤50%, P = 0.81, $I^2$ = 0%), the result of two studies [30, 33] showed that postpartum rest time less than 3 months was identified as a risk factor for LM (OR 4.71, 95%CI [3.92, 5.65]). The forest plot is shown in Fig 6.

There was obvious heterogeneity among the following risk factors ($I^2$>50%): the first six months postpartum (P<0.00001, $I^2$ = 93%), postpartum prone sleeping position (P<0.0006, $I^2$ = 80%) and postpartum mood disorders (P<0.00001, $I^2$ = 83%), the data of these factors were pooled using the random effect model. The results showed that the first six months postpartum [29, 30, 32, 34, 35] (OR 5.11, 95%CI [2.66, 9.82]), postpartum prone sleeping position [29–31, 35, 37] (OR 2.46, 95%CI [1.58, 3.84]) and postpartum mood disorders [29–31, 33–35, 37–39] (OR 1.47, 95%CI [1.06, 2.02]) were identified as the significant risk factors for LM. The forest plots are shown in Fig 7. Similarly, the result from one cross-sectional study [41]

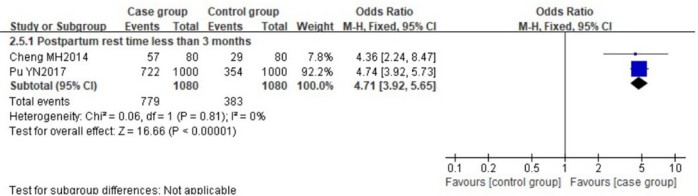

**Fig 6. Forest plot comparing the pooled risk of postpartum rest time less than 3 months.**

involving 68 participants reported that mother with the prone sleeping position experienced a higher risk of LM (OR 2.26, 95%CI [1.23, 4.11]).

## 3.4 Sensitivity analysis

Sensitivity analysis was performed by eliminating each study one by one, at a time the summary P values and ORs of the remaining studies were recalculated. The results of each breast-feeding duration > 0.5 h, nipple sucking and postpartum mood disorders partially deviated from the 95% confidence interval estimated by meta-analysis, indicating that the robustness of the currently available data for these factors was relatively poor. The pooled results of these risk factors may be influenced by high-risk bias studies (Figs 8–10). The robustness of meta-analysis for other risk factors is acceptable.

## 3.5 The analysis of PAR and Nfs

The PAR of risk factors (OR>1) significantly associated with LM were calculated in this study. The PAR for the first six months postpartum had the highest chance of exposure (65.93%), followed by improper milking method (59.14%), postpartum rest time less than 3 months (56.95%), repeated milk stasis (49.75%) and abnormal nipple or crater nipple (42.05%). The PAR of the history of diabetes (6.8%) demonstrated a relatively low chance of exposure in this population level. The results of PAR for all risk factors were shown in Table 2.

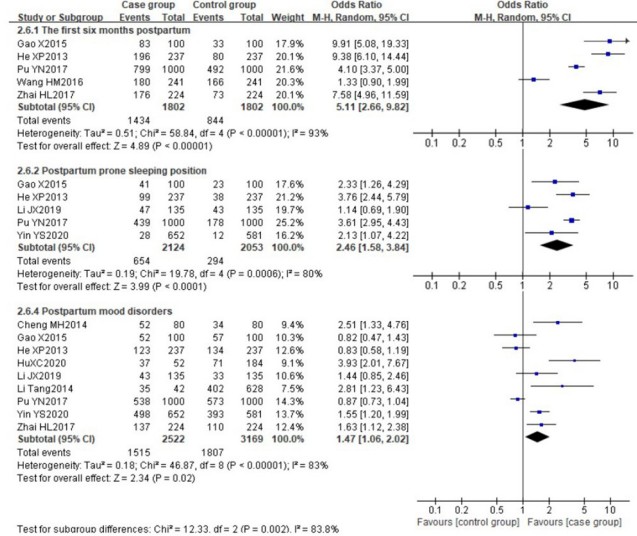

**Fig 7. Forest plot comparing the pooled risks of the first six months postpartum, postpartum prone sleeping position and postpartum mood disorders.**

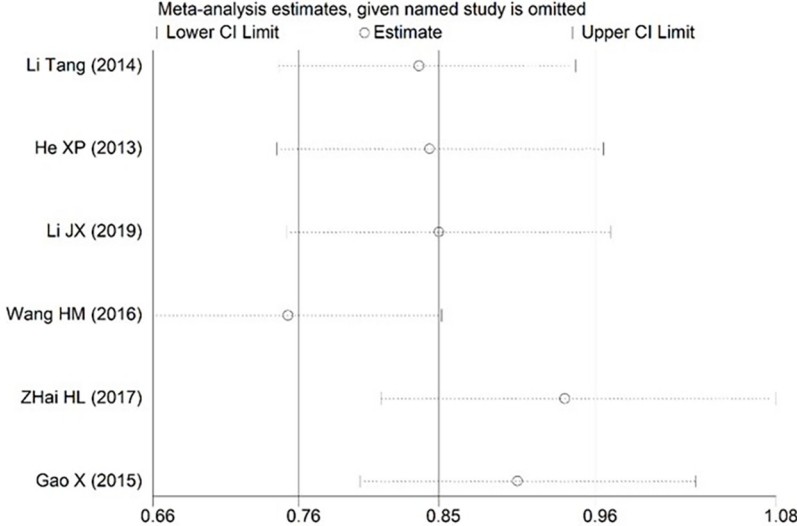

**Fig 8. Sensitivity analysis for each breastfeeding duration>0.5 h.**

Nfs estimates of the risk factors were created with the formula obtained from the data analysis section. Nfs estimates for all risk factors illustrated that there was relatively good robustness of the pooled results. Studies required to nullify the current results were relatively higher based on findings of Nfs estimates (P = 0.05), such as the Nfs0.05 for history of diabetes was n = 16, while higher numbers were required to nullify the effect in breast trauma (n = 186) and in little or no nipple cleaning (n = 178). These also indicated that the publication bias may not exist [27, 42]. The results of Nfs for all risk factors were shown in Table 2.

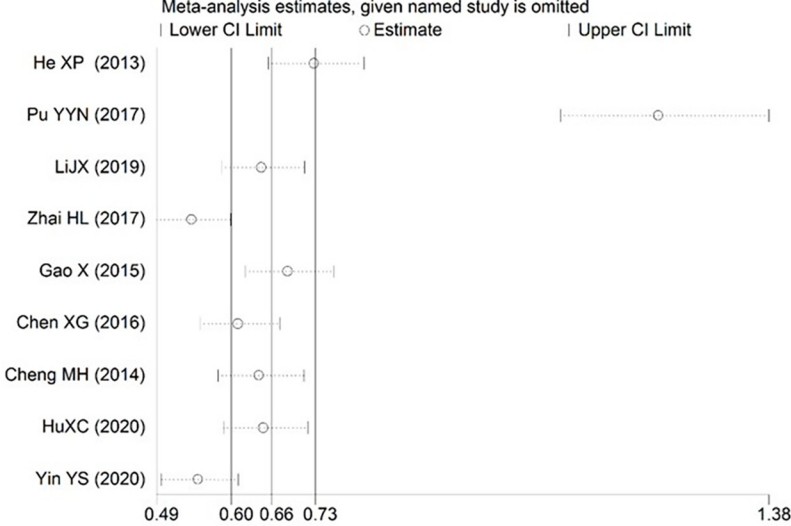

**Fig 9. Sensitivity analysis for nipple sucking.**

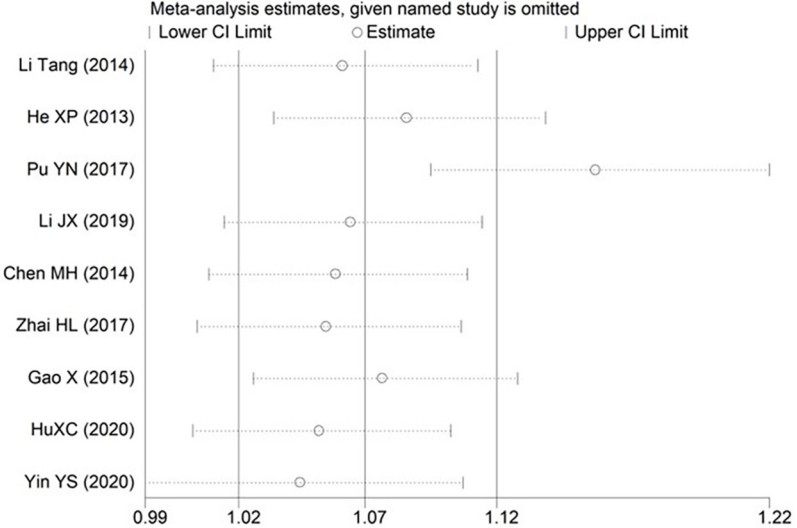

**Fig 10. Sensitivity analysis for postpartum mood disorders.**

### 3.6 Publication bias

Additionally, Egger's linear regression analysis was based on studies that reported risk factors for little or no nipple cleaning and postpartum mood disorders. Egger's publication bias plots

**Table 2. The results of the PAR and the Nfs for risk factors associated with LM.**

| Risk factor | Study | The population attributable risk percent | | | The fail-safe number | |
|---|---|---|---|---|---|---|
| | | OR | Pm (%) | PAR (%) | Nfs0.05 | Nfs0.01 |
| *1 Risk factors related to puerperium behaviors and characteristics* | | | | | | |
| Improper breastfeeding posture | 3 | 2.47 | 24.54 | 26.52 | 32 | 13 |
| Repeated milk stasis | 3 | 6.23 | 18.93 | 49.75 | 26 | 12 |
| Improper milking method | 2 | 6.79 | 25.00 | 59.14 | NA | NA |
| Little or no nipple cleaning | 9 | 2.05 | 31.08 | 24.73 | 178 | 84 |
| *2 Risk factors related to maternal characteristics* | | | | | | |
| Cesarean section | 4 | 1.51 | 44.51 | 18.61 | 28 | 12 |
| Breast massage experience of non-medical staff | 3 | 1.51 | 35.43 | 15.31 | 21 | 9 |
| History of diabetes | 3 | 2.26 | 5.79 | 6.81 | 16 | 6 |
| History of mastitis | 3 | 2.36 | 9.55 | 11.60 | 19 | 7 |
| Abnormal nipple or crater nipple | 6 | 3.94 | 24.46 | 42.05 | 119 | 56 |
| Primipara | 6 | 1.73 | 65.69 | 32.62 | 98 | 47 |
| Breast trauma | 7 | 3.07 | 9.11 | 15.98 | 186 | 88 |
| Low education level | 5 | 1.63 | 48.06 | 23.29 | 39 | 17 |
| *3 Risk factors related to postpartum period* | | | | | | |
| The first six months postpartum | 5 | 5.11 | 46.84 | 65.93 | 72 | 33 |
| Postpartum prone sleeping position | 5 | 2.46 | 14.32 | 17.42 | 57 | 25 |
| Postpartum mood disorders | 9 | 1.47 | 57.02 | 21.27 | 147 | 68 |
| Postpartum rest time less than 3 months | 2 | 4.71 | 35.46 | 56.95 | NA | NA |

Note: Pm is an estimate of the population prevalence of that risk factor derived from the control group based on meta-analyses; 'Pm' is expressed as an approximation of 'Pe' as the prevalence of exposure in the population. NA: not available; PAR: the population attributable risks percent; OR: odds ratio; Nfs: fail-safe number.

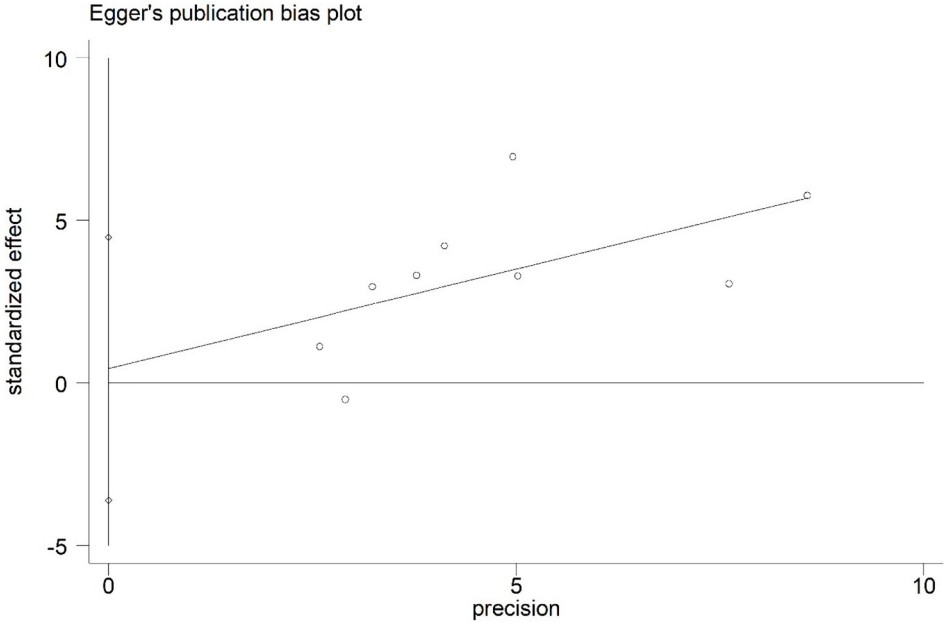

**Fig 11. Egger's publication bias plot for little or no nipple cleaning.**

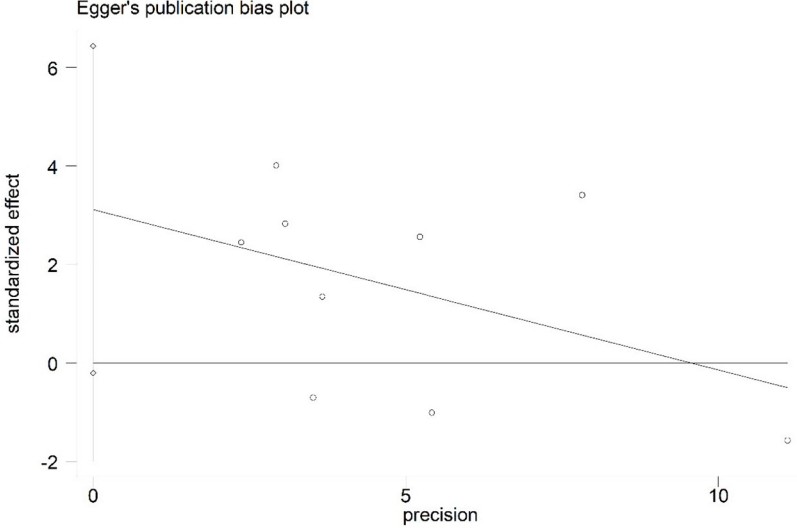

**Fig 12. Egger's publication bias plot for postpartum mood disorders.**

of little or no nipple cleaning (Std. Err = 1.71, t = 0.25, P = 0.807) and postpartum mood disorders (Std. Err = 1.41, t = 2.21, P = 0.062) are shown in Figs 11 and 12. The results of Egger's test demonstrated that the included studies may have no statistically significant publication bias (P>0.05).

## 4 Discussions

LM has a serious physiological and psychological effect on breastfeeding women [43]. In previous systematic reviews, many factors related to LM in the world were studied [17]. However,

the findings were only synthesized narratively due to the large population differences and significant heterogeneity in these studies, and only one study from China was included. In order to obtain more meaningful data on the significant risk factors of LM in China, we conducted a comprehensive meta-analysis based on published studies involved Chinese women only. Fourteen studies were included in this review, involving a total of 8032 participants. Numerous risk factors were assessed and the following risk factors were identified as significant risk factors for LM. (1) Risk factors related to puerperium behaviors and characteristics: improper breastfeeding posture, repeated milk stasis, improper milking method, and little or no nipple cleaning before breastfeeding. (2) Risk factors related to maternal characteristics: cesarean section, breast massage experience of non-medical staff, abnormal nipple or crater nipple, primipara, breast trauma and low education level (bachelor below). (3) Risk factors associated with the postpartum period: postpartum rest time less than 3 months, the first six months postpartum, postpartum prone sleeping position and postpartum mood disorders.

Similar to previous studies [17, 44], LM was associated with risk factors such as abnormal nipple or cratered nipple, history of mastitis, breast trauma and postpartum mood disorders (such as stress, anxiety, irritability and confusion). Most importantly, many of the risk factors mentioned in this review seemed amenable to mediation by mother's breastfeeding behavior and practices. In other word, by controlling some of the modifiable risk factors [10, 17, 37] such as improper breastfeeding posture, improper milking method, milk stasis, nipple cleaning condition, breast massage experience of non-medical staff and postpartum sleeping posture, the incidence of LM may be reduced. In particular, breastfeeding women can control some of the risk factors by themselves. On the other hand, the important risk factors to the initiation of LM can be different due to variation in the populations, socioeconomic status and cultural background [11, 17]. For example, in present study, primipara was identified as a risk factor for LM, while in the study from U.S [45], the expected association between mastitis and primipara was not found. Therefore, some results in this review may not be applicable to people in other countries or regions.

In order to estimate the potential impact of risk factors on LM at the population level and better guide clinical practice, we calculated PAR percent for the risk factors significantly associated with LM. In particular, mothers with a history of diabetes were reported to be associated with LM [29, 32, 33]. However, since the chance of exposure (PAR = 6.81%) seemed relatively low in this population level and only three studies [29, 32, 33] were included in the analysis, further studies are needed to confirm this finding. In addition to the history of diabetes, the finding in present study demonstrated that risk factors related to puerperium behaviors and characteristics, maternal characteristics and postpartum period had potential negative impact on the incidence of LM in Chinese women with PAR estimates ranging from 11.6%6 to 65.93%.

In this study, the result found that the prolonged breastfeeding (each breastfeeding duration > 0.5 h) was not an independent risk factor for LM, which was inconsistent with previous studies [32, 39]. In other hand, the sensitivity analysis of this result indicated that the robustness of the currently available data was relatively poor. Therefore, further studies on this topic are recommended to confirm whether it is a risk factor related to LM.

Previous research has found that preterm infants have an immature sucking behavior, which may have an influence on the capacity of exclusively breastfed for a period of weeks or months [46]. Similar to previous studies [31, 34, 36], sucking manner of infants (nipple sucking) was found to be another risk factor associated with LM. In addition, the Department of Maternal and Child, China's Ministry of Health has issued a breastfeeding manual, which encourages the women to help the infants to suck nipples and areola during breastfeeding [29]. However, the results of this review showed that there was no statistically significant

relationship between sucking manners and LM, so it was not possible to determine the effect of nipple sucking, and further studies were recommended.

The results of our study demonstrated that there was a link between postpartum mood disorders and LM. Similar to a previous study [47], maternal mood disorders were identified as the risk factor associated with LM. Besides, it was reported that negative emotions can reduce the body's "SIgA" level and change the biochemistry of both the local organ microenvironment as well as the global systemic inflammatory burden, which will lead to a decline in the body's resistance to some diseases [48, 49]. However, sensitivity analysis in this review revealed that the robustness of the currently available data for postpartum mood disorders was not good, which may be related to the inconsistent severity and definition of mood disorders in different studies. Accordingly, we believed that postpartum mood disorders were the important risk factor for LM, practitioners should be aware of the possibility of LM in mothers with any mood disorder, especially those with a history of mental health problems [37, 50].

The findings of this review might provide evidence-based information for the high-risk factors of LM in China, which will be helpful for the multidisciplinary team or practitioners involved in maternal and infant breastfeeding management to provide appropriate management advice, scientific treatment strategies and effectively individual care. Most importantly, this review provides a reference for the prevention of LM and further study on the pathogenic factors of LM. There is no denying that this study has some limitations. Firstly, the disparities in heterogeneity among studies may have affected the effectiveness of statistical analysis, due to potential confounding factors such as sample size, design differences, underlying population characteristics, etc. Secondly, the effect estimate could not be calculated for all risk factors, because more than two studies related to the same defined risk factor for LM were summarized in the meta-analysis. Finally, this review included literature mainly from Chinese mainland and the included studies involved Chinese women only, which may restrict the generalizability and interpretation of the findings. However, our findings made an important contribution to determining the well-accepted risk factors related to LM by integrating studies involving LM risk factors and specified the aspects that need to be investigated in the future.

## 5 Conclusions

The significant risk factors for LM were improper milking method, repeated milk stasis, the first six months postpartum, postpartum rest time less than 3 months, abnormal nipple or crater nipple, breast trauma, improper breastfeeding posture, postpartum prone sleeping position, little or no nipple cleaning, primipara, low education level, cesarean section, breast massage experience of non-medical staff and postpartum mood disorders. These findings have some reference value for the prevention, treatment and individual care of LM. In particular, the incidence of LM can be reduced by controlling some of the modifiable risk factors.

## Supporting information

**S1 File. PRISMA 2009 checklist.**
(DOC)

**S2 File. The protocol of this review.**
(PDF)

**S3 File. Search terms in PubMed.**
(DOCX)

**S4 File. Summary of abbreviations in text.**
(DOCX)

**S1 Table. Study quality of case-control studies.**
(DOCX)

**S2 Table. Study quality of cohort studies.**
(DOCX)

**S3 Table. Study quality of cross-sectional studies.**
(DOCX)

## Author Contributions

**Conceptualization:** Xiao-Hua Pei.

**Data curation:** Ai-Jing Chu, Shi-Bing Liang, Li-Yan Jia.

**Formal analysis:** Bao-Yong Lai, Ai-Jing Chu.

**Funding acquisition:** Ying-Yi Fan, Xiao-Hua Pei.

**Investigation:** Ai-Jing Chu, Shi-Bing Liang.

**Methodology:** Bo-Wen Yu, Shi-Bing Liang.

**Software:** Li-Yan Jia.

**Writing – original draft:** Bao-Yong Lai.

**Writing – review & editing:** Bao-Yong Lai, Bo-Wen Yu, Jian-Ping Liu, Ying-Yi Fan, Xiao-Hua Pei.

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
