## [Decision Letter · Decision Letter 0]

24 Feb 2021

PONE-D-20-36977

Risk factors for lactation mastitis in China: a systematic review and meta-analysis

PLOS ONE

Dear Dr. Lai,

Thank you for submitting your manuscript to PLOS ONE. After careful consideration, we feel that it has merit but does not fully meet PLOS ONE’s publication criteria as it currently stands. Therefore, we invite you to submit a revised version of the manuscript that addresses the points raised during the review process.

ACADEMIC EDITOR comments:

Dear authors!

Your article deals with a clinically relevant topic and I am in favor of publishing it. Two experts in the field have reviewed your work and note the need for major revisions. Besides the need to improve readability by streamlining the text, some english revisions (there are some non-technical terms such as bad mood instead of depression etc) are required. Further, you should elaborate in your discussion on the fact that only Chinese women were investigated, the rationale for doing so and discuss how the results may or not be applicable to other populations.

We look forward to receiving your revised manuscript.

Kind regards,

Pascal A. T. Baltzer, M.D.

Academic Editor

PLOS ONE

2. Please ensure that you refer to Figure 12 in your text as, if accepted, production will need this reference to link the reader to the figure.

3. Please include your tables as part of your main manuscript and remove the individual files. Please note that supplementary tables (should remain/ be uploaded) as separate "supporting information" files

Reviewers' comments:

Reviewer's Responses to Questions

**Comments to the Author**

1. Is the manuscript technically sound, and do the data support the conclusions?

Reviewer #1: Yes

Reviewer #2: Yes

2. Has the statistical analysis been performed appropriately and rigorously? 

Reviewer #1: Yes

Reviewer #2: Yes

3. Have the authors made all data underlying the findings in their manuscript fully available?

Reviewer #1: No

Reviewer #2: Yes

4. Is the manuscript presented in an intelligible fashion and written in standard English?

Reviewer #1: Yes

Reviewer #2: Yes

5. Review Comments to the Author

Reviewer #1: The study is certainly interesting and has a clear clinical but also social impact. The real limitation I see is the fact that the authors have chosen to include Chinese patients only. I do not understand in primis the reason for this choice and, moreover, I do not completely understand how they selected these patients during the research.

Light English revision is needed.

Minor concerns:

1. Abstract:

a. Why using the term “inconsistent”? Please comment.

b. I would suggest avoiding “This is the first meta-analysis…”, you cannot know someone published one during your revisions, even if not registered in PROSPERO.

c. Using the term “pooled risk factors” is inappropriate. A pooled risk is a continuous variable, risk factors can be categorical too.

d. Please follow a rationale presenting the included risk factors, for example from the one with the highest OR.

2. Keywords: OK

3. Introduction:

a. “In addition, an extensive search of Chinese and English literature has not found any quantitative meta-analysis to assess the risk factors associated with LM in China”: as in the abstract.

b. From the title it is understood that the authors want to include only studies on Chinese patients, but from the introduction this is not clear.

4. Methods:

a. PRISMA ref should be referenced in the text.

b. “All considered participants were Chinese women…”: this is not so clear to me. How were studies with mixed populations managed?

c. In the search string I see at least two issues: “Lactation Mastitis” is NOT a MeSH term; in the whole search string I cannot see the term “Chinese”. How did the authors include only studies with Chinese patients with this type of research?

d. “Two authors independently selected the studies and extracted the detailed data of the eligible trials”: using the term “trial” as a synonym of “study” is highly debatable.

e. In the methods it is not advisable using the term “etc.”, they should be used to replicate the study. You can use a table.

f. Line 138: add a ref.

g. “Any disagreements” -> any disagreement

h. Despite the use of I2 heterogeneity to guide the use of random or fixed effect method is very diffuse, it is highly debatable. I could accept it, but this choice should be supported by appropriate references.

5. Results:

a. Line 166: “that were included” -> “AND were included”.

b. Line 171 “of which” can be removed.

c. Line 176: “acceptable” has not been defined in the methods.

d. Line 176: provide the number instead of “Most of…”

e. “Results of meta-analysis”: using the term “including” while reporting the entire list of included risk factors is inappropriate, "including" should be used when only a fraction is listed.

f. Line 202: at least the I2 value should be reported.

g. Line 209: “pooled risk” is better than “pooled results”.

h. Line 214 and 242: please rephrase, after listing all the risk factors you start with “we used ...”

i. Line 289 and 291: “another” -> “other”

6. Discussion:

a. In the first paragraph the authors should specify they included published studies including Chinese women only.

b. Line 355: I think the manual encourages the women to help the infants, not directly the infants.

7. Tables:

a. Table 1: why using “T” for cases? It is not immediate for the reader.

b. Table 1: please revise the first column, in some IDs I see spaces between first author and year of publication, in other cases no space has been inserted.

c. Table 1: why reporting controls in a cohort study?

d. Table 1: the column of the risk factors can be removed; it is impossible to understand.

e. Table 2 title should be enlarged to help understanding the content.

8. Figures:

a. X axis of the forest plots should be duplicated to better interpretate the first plot (for example, improper breastfeeding posture for Fig.2)

b. In the first line, “case group” and “control group” should be capitalized.

Reviewer #2: Dear authors, I have reviewed your manuscript entiteld: “Risk factors for lactation mastitis in China: a systematic review and meta-analysis“ and have the following comments:

Abstract:

Please structure your abstract into i.e. Background, Purpose , Material and Methods, Results and Conclusion, or a similar subdivision for better readability.

First line: use present tense.

electronic literature databases - please identify, search terms are missing, reference to study quality control is missing

please rephrase “postpartum bad mood”

Introduction:

- Line 70: “proabably” should be rephrased

- stick to one tense (don’t switch between past an present tense)

- “The World Health Organization recommends that infants start breastfeeding within one hour of life, are exclusively breastfed for six months, with timely introduction of adequate, safe and properly fed complementary foods while continuing breastfeeding for up to two years of age or beyond.“ – your statement takes this WHO recommendation out of context – please rephrase. In particular it means that the WHO recommends exclusive breastfeeding for the first 6 months and for up to two years and beyond. It does not simply recommend it for the first 6 months as you stated.

- Line 83 – please add reference to this statement

- Line 89: in order to prolong lacation

- Line 96: is there one for caucasian women or hispanic women or black african women?

- Last line of introduction is more of a conclusion – please rephrase – this way it does not fit the introduction

M&M:

- Extracted items – please name all, don’t say “etc.”

- appropriate choice of quality assessment tools

- Figure 1 – remove typo (records excluded not recorders, same mistake in the box below), specify other non-conforming studies (I find 40 to be a large number of random non-conforming studies), studies included in qualitative synthesis n=30 – change to 14

Results:

- Line 189: rephrase – unclear

- Line 200: Postpartum bad mood should be replaced by a more scientific term

- 202: no significant heterogeneity (please provide statistical measure to back this assessment)

- Paragraph 2.5 The analysis of Nfs and PAR is confusing – you only focus on two risk factors in the text and cite the rest in the table – please balance your reporting

- Publication bias: Egger’s test finds no statistically significant results – yet you state that “The results of Egger's test demonstrated that the included studies may have potential publication bias.“ Please explain.

Discussion:

- Line 316: Lacks reference

- Line 338: add reference

- Line 339 (add reference to the 3 studies)

- Line 342 (add reference to the 3 studies)

- 342-343: define other risk factors and provide references to the studies that provide the background data

General: While the english narrative is mostly good, the manuscript suffers from changes between present tense and past tense and is extremely lengthy – it needs to be streamlined and more focused. Editing by an English native familiar with the topic is recommended.

Kind regards,

Reviewer 2

6. PLOS authors have the option to publish the peer review history of their article (what does this mean?). If published, this will include your full peer review and any attached files.

Reviewer #1: No

Reviewer #2: No

---

## [Author Response · Author response to Decision Letter 0]

2 Apr 2021

Author response to comments from academic editor and reviewers

Comments from academic editor:

Dear authors!

Your article deals with a clinically relevant topic and I am in favor of publishing it. Two experts in the field have reviewed your work and note the need for major revisions. Besides the need to improve readability by streamlining the text, some english revisions (there are some non-technical terms such as bad mood instead of depression etc) are required. Further, you should elaborate in your discussion on the fact that only Chinese women were investigated, the rationale for doing so and discuss how the results may or not be applicable to other populations.

Author reply:

Dear academic editor, 

Many thanks for your helpful comments and suggestion. We have revised the manuscript accordingly and the paper was carefully edited again. Please see English editing throughout the manuscript.

Comments from reviewers:

-Reviewer #1: 

Q1: The study is certainly interesting and has a clear clinical but also social impact. The real limitation I see is the fact that the authors have chosen to include Chinese patients only. I do not understand in primis the reason for this choice and, moreover, I do not completely understand how they selected these patients during the research.

Author reply: Thank you for your comment. This review summarized the current evidence of risk factors for lactation mastitis (LM) in China and numerous risk factors were assessed. Considering the complexity of LM etiology and variation in the populations, this review included studies involved Chinese participants only. We agreed with the issues you mentioned above, we have added your concern as the limitation of this review. Additionally, we have added information on how the participants were selected during the research in our revised manuscript and you can see“2.2 Study characteristics and quality assessment of included studies” section.

Q2: Light English revision is needed.

Minor concerns:

1. Abstract:

a. Why using the term “inconsistent”? Please comment.

Author reply: Many thanks for your comments. We have reedited this term and added comments for it.

b. I would suggest avoiding “This is the first meta-analysis…”, you cannot know someone published one during your revisions, even if not registered in PROSPERO.

Author reply: Thank you very much. We agreed with you and have rephrased this sentence.

c. Using the term “pooled risk factors” is inappropriate. A pooled risk is a continuous variable, risk factors can be categorical too.

Author reply: We agreed with you and have rephrased the term “pooled risk factors”.

d. Please follow a rationale presenting the included risk factors, for example from the one with the highest OR.

Author reply: Thanks. We agreed with you and have shown the included risk factors from the one with the highest OR.

2. Keywords: OK

Author reply: Many thanks for your comments.

3. Introduction:

a. “In addition, an extensive search of Chinese and English literature has not found any quantitative meta-analysis to assess the risk factors associated with LM in China”: as in the abstract.

Author reply: Thank you very much. We agreed with you and have rephrased this sentence.

b. From the title it is understood that the authors want to include only studies on Chinese patients, but from the introduction this is not clear.

Author reply: Many thanks for your comments. We have carefully edited the introduction section again. Please see revisions throughout manuscript.

4. Methods:

a. PRISMA ref should be referenced in the text.

Author reply: Thank you very much, we have added PRISMA ref the in the text. Please see reference [18].

b. “All considered participants were Chinese women…”: this is not so clear to me. How were studies with mixed populations managed?

Author reply: Many thanks for your comments. We have reedited this sentence. The modification now is “All considered participants were Chinese women. If the studies are mixed populations, data from Chinese women could be analyzed separately regardless of their age or race”.

c. In the search string I see at least two issues: “Lactation Mastitis” is NOT a MeSH term; in the whole search string I cannot see the term “Chinese”. How did the authors include only studies with Chinese patients with this type of research?

Author reply: Thank you for your concern. We have revised and added search terms. In addition, we retrieved the database again based on the modified search terms.

d. “Two authors independently selected the studies and extracted the detailed data of the eligible trials”: using the term “trial” as a synonym of “study” is highly debatable.

Author reply: Thank you for your concern. The modification now is “Two authors independently selected the studies and extracted the detailed data of the eligible studies” and we have avoided using the term “trial” as a synonym of “study” in our study.

e. In the methods it is not advisable using the term “etc.”, they should be used to replicate the study. You can use a table.

Author reply: Thanks. We agreed with you and have named all items in the methods.

f. Line 138: add a ref.

Author reply: Thanks. We have added a ref in the text. Please see reference [24].

g. “Any disagreements” -> any disagreement

Author reply: Thanks. We have revised this.

h. Despite the use of I2 heterogeneity to guide the use of random or fixed effect method is very diffuse, it is highly debatable. I could accept it, but this choice should be supported by appropriate references.

Author reply: Thank you for your advice. We have added a ref in the text. Please see reference [25].

5. Results:

a. Line 166: “that were included” -> “AND were included”.

Author reply: Thanks. We have revised this.

b. Line 171 “of which” can be removed.

Author reply: Thanks. We have removed it.

c. Line 176: “acceptable” has not been defined in the methods.

Author reply: Many thanks for your comments. We have added the definition of “acceptable quality” in 1.4 Quality assessment section. 

d. Line 176: provide the number instead of “Most of…”

Author reply: Thanks. We have revised it.

e. “Results of meta-analysis”: using the term “including” while reporting the entire list of included risk factors is inappropriate, "including" should be used when only a fraction is listed.

Author reply: Many thanks for your comments and we have revised this.

f. Line 202: at least the I2 value should be reported.

Author reply: Thank you for your advice. We have added I2 value and P value in the corresponding results.

g. Line 209: “pooled risk” is better than “pooled results”.

Author reply: Thanks. We agreed with you and have revised this.

h. Line 214 and 242: please rephrase, after listing all the risk factors you start with “we used ...”

Author reply: Many thanks for your comments. We have rephrased these sentences.

i. Line 289 and 291: “another” -> “other”

Author reply: Thanks. We have revised it.

6. Discussion:

a. In the first paragraph the authors should specify they included published studies including Chinese women only.

Author reply: We agreed with you and have added “published studies including Chinese women only” in discussion. The modification now is “…. we conducted a comprehensive meta-analysis based on published studies involved Chinese women only.”

b. Line 355: I think the manual encourages the women to help the infants, not directly the infants.

Author reply: Thanks. We have revised it.

7.Tables:

a. Table 1: why using “T” for cases? It is not immediate for the reader.

Author reply: Many thanks for your comments. We have rephrased them in Table 1.

b. Table 1: please revise the first column, in some IDs I see spaces between first author and year of publication, in other cases no space has been inserted.

Author reply: Thanks. We have reedited these.

c. Table 1: why reporting controls in a cohort study?

Author reply: Thanks. We have reedited it.

d. Table 1: the column of the risk factors can be removed; it is impossible to understand.

Author reply: We agreed with you and have removed “the column of the risk factors” in Table 1.

e. Table 2 title should be enlarged to help understanding the content.

Author reply: Thanks. We have enlarged the title of Table 2. The modification now is “Table 2 the results of the PAR and the Nfs for risk factors associated with LM”.

8. Figures:

a. X axis of the forest plots should be duplicated to better interpretate the first plot (for example, improper breastfeeding posture for Fig.2)

Author reply: We agreed with you and have duplicated X axis of the forest plots in according figures.

b. In the first line, “case group” and “control group” should be capitalized.

Author reply: Thanks. We have capitalized them in according figures.

-Reviewer #2: 

Dear authors, I have reviewed your manuscript entitled: “Risk factors for lactation mastitis in China: a systematic review and meta-analysis” and have the following comments:

1. Abstract:

a. Please structure your abstract into i.e. Background, Purpose , Material and Methods, Results and Conclusion, or a similar subdivision for better readability.

Author reply: We really appreciated your advice. We have reedited the abstract accordingly.

b. First line: use present tense.

Author reply: Thanks. We have reedited this.

c. electronic literature databases - please identify, search terms are missing, reference to study quality control is missing

Author reply: Many thanks for your comments. We have added information on electronic literature databases and study quality control. Considering the limitation of the number of words in abstract section, we only presented the search terms in the full text.

d. please rephrase “postpartum bad mood”

Author reply: Many thanks for your comments. We have rephrased “postpartum bad mood” and the modification now is “postpartum mood disorders”.

2 Introduction:

a. Line 70: “proabably” should be rephrased

Author reply: Many thanks for your comments. We have rephrased “proabably” as “generally”.

b. stick to one tense (don’t switch between past an present tense)

Author reply: Thanks. we have carefully reedited manuscript again.

c. “The World Health Organization recommends that infants start breastfeeding within one hour of life, are exclusively breastfed for six months, with timely introduction of adequate, safe and properly fed complementary foods while continuing breastfeeding for up to two years of age or beyond.“ – your statement takes this WHO recommendation out of context – please rephrase. In particular it means that the WHO recommends exclusive breastfeeding for the first 6 months and for up to two years and beyond. It does not simply recommend it for the first 6 months as you stated.

Author reply: Many thanks for your comments. we have rephrased this sentence again and the modification now is “The World Health Organization (WHO) or international guidelines highly recommends that infants are exclusively breastfed for the first six months of life and continue breastfeeding for up to two years of age or older, because breastfeeding can provide the best nutritional start for infants [7, 8]”.

d. Line 83 – please add reference to this statement

Author reply: Thanks. We have added the reference to it. Please see reference [9].

e. Line 89: in order to prolong lacation

Author reply: Thanks. We have revised it.

f. Line 96: is there one for caucasian women or hispanic women or black african women?

Author reply: Many thanks for your comments. We have revised this sentence and one recent systematic review [17] published in 2020 summarized the evidence on risk factors for LM globally--(twenty-six articles were included, 10 (38%) were conducted in Australia or New Zealand, seven (27%) in Europe (one each in Finland, Denmark, Spain, Sweden, and Germany, and two in the United Kingdom), four (15%) in the United States, three (12%) in Asia (Nepal, China, and Iran), and two (8%) in Africa (Gambia and Ghana)). However, the effect size of risk factors was not finally pooled due to methodological differences between these studies and only one study from China was included.

g. Last line of introduction is more of a conclusion – please rephrase – this way it does not fit the introduction

Author reply: We agreed with you and have removed them from introduction section.

2 M&M:

a. - Extracted items – please name all, don’t say “etc.”

Author reply: We agreed with you and have named all items.

b.- appropriate choice of quality assessment tools

Author reply: Many thanks for your comments. In this review, the quality of the case-control study and cohort study was respectively assessed according to the criteria of the Newcastle-Ottawa Scale (NOS) [19], and the quality of cross-sectional study was evaluated using the modified NOS [20, 21] based on the recommendation in other studies.

c.- Figure 1 – remove typo (records excluded not recorders, same mistake in the box below), specify other non-conforming studies (I find 40 to be a large number of random non-conforming studies), studies included in qualitative synthesis n=30 – change to 14

Author reply: Thanks. We have revised all issues your mentioned above.

3 Results:

a. - Line 189: rephrase – unclear

Author reply: Many thanks for your comments. We have rephrased this sentence and the modification now is “More than two studies involving the same defined risk factor for LM were summarized in the meta-analysis”.

b. - Line 200: Postpartum bad mood should be replaced by a more scientific term 

Author reply: Thanks. We have rephrased this term and the modification now is “postpartum mood disorders”.

c. - 202: no significant heterogeneity (please provide statistical measure to back this assessment)

Author reply: Many thanks for your comments. We have added I2 value and P value in the corresponding results.

d. - Paragraph 2.5 The analysis of Nfs and PAR is confusing – you only focus on two risk factors in the text and cite the rest in the table – please balance your reporting

Author reply: Many thanks for your comments. We have summarized the results in “2.5 The analysis of Nfs and PAR” section again.

e. - Publication bias: Egger’s test finds no statistically significant results – yet you state that “The results of Egger's test demonstrated that the included studies may have potential publication bias.“ Please explain.

Author reply: We agreed with you and have rephrased this result. The modification now is “The results of Egger's test demonstrated that the included studies may have no statistically significant publication bias (P＞0.05)”.

4.Discussion:

a. - Line 316: Lacks reference

Author reply: Thanks. We have added the reference to it. Please see reference [43].

b. - Line 338: add reference

- Line 339 (add reference to the 3 studies)

Author reply: Thanks. We have added the reference to them. Please see reference [29, 32, 33].

c. - Line 342 (add reference to the 3 studies)

Author reply: Thanks. We have removed this sentence and have rephrased this sentence.

d. - 342-343: define other risk factors and provide references to the studies that provide the background data

Author reply: Many thanks for your comments. We have rephrased this sentence.

5. General: While the english narrative is mostly good, the manuscript suffers from changes between present tense and past tense and is extremely lengthy – it needs to be streamlined and more focused. Editing by an English native familiar with the topic is recommended.

Author reply: Thanks for your concerns. We agreed with the issues you mentioned above and we have carefully edited the paper again. 

Thank you again for your time and effort.

With best regards,

Sincerely yours,

Bao-Yong Lai 

Third Affiliated Hospital of Beijing University of Chinese Medicine, 

Beijing100029, China.

Email: by_lai@126.com; baoyonglai@bucm.edu.cn

Xiao-Hua Pei, MD, PhD

Professor, Third Affiliated Hospital of Beijing University of Chinese Medicine, 

Beijing University of Chinese Medicine, Beijing 100029, China

Email: pxh_127@163.com

---

## [Editor Report · Decision Letter 1]

22 Apr 2021

Risk factors for lactation mastitis in China: a systematic review and meta-analysis

PONE-D-20-36977R1

Dear Dr. Lai,

We’re pleased to inform you that after refviewing your revised manuscript, your paper has been judged scientifically suitable for publication and will be formally accepted for publication once it meets all outstanding technical requirements.

Kind regards,

Pascal A. T. Baltzer, M.D.

Academic Editor

PLOS ONE

---

## [Editor Report · Acceptance letter]

30 Apr 2021

PONE-D-20-36977R1 

Risk factors for lactation mastitis in China: a systematic review and meta-analysis 

Dear Dr. Lai:

I'm pleased to inform you that your manuscript has been deemed suitable for publication in PLOS ONE. Congratulations! Your manuscript is now with our production department. 

Kind regards, 

on behalf of

Dr. Pascal A. T. Baltzer 

Academic Editor

PLOS ONE